# Complete structure of the bacterial flagellar hook reveals extensive set of stabilizing interactions

Hideyuki Matsunami[1], Clive S. Barker[1], Young-Ho Yoon[1], Matthias Wolf[2] & Fadel A. Samatey[1]

The bacterial flagellar hook is a tubular helical structure made by the polymerization of multiple copies of a protein, FlgE. Here we report the structure of the hook from *Campylobacter jejuni* by cryo-electron microscopy at a resolution of 3.5 Å. On the basis of this structure, we show that the hook is stabilized by intricate inter-molecular interactions between FlgE molecules. Extra domains in FlgE, found only in *Campylobacter* and in related bacteria, bring more stability and robustness to the hook. Functional experiments suggest that *Campylobacter* requires an unusually strong hook to swim without its flagella being torn off. This structure reveals details of the quaternary organization of the hook that consists of 11 protofilaments. Previous study of the flagellar filament of *Campylobacter* by electron microscopy showed its quaternary structure made of seven protofilaments. Therefore, this study puts in evidence the difference between the quaternary structures of a bacterial filament and its hook.

[1] Trans-Membrane Trafficking Unit, Okinawa Institute of Science and Technology Graduate University, 1919-1 Tancha, Onna, Kunigami 904-0495, Japan. [2] Molecular Cryo-Electron Microscopy Unit, Okinawa Institute of Science and Technology Graduate University, 1919-1 Tancha, Onna, Kunigami 904-0495, Japan. Correspondence and requests for materials should be addressed to F.A.S. (email: bakary.macy.samatey@gmail.com).

Bacterial flagella have long been studied[1–5], but many aspects of their structure and function are still eluding us. The structure of the hook can be described as a tubular helical structure[6]. The hook functions like a universal joint. It transmits the torque, produced by the motor located in the cell membrane, to the filament that acts like a propeller[7]. The assembly of about a 100 copies of a single protein, FlgE, makes the bacterial flagellar hook. An exported molecular ruler protein, FliK, controls the hook length[8]. Cells bearing mutations in the *fliK* gene produce abnormally long 'polyhook' structures[9]. The bacterial flagellar motor rotates at frequencies that vary between 100 Hz and 2,000 Hz depending on bacterial species[10]. The hook undergoes multiple conformational changes while rotating around its axis[6,11,12]. During these conformational changes, the interactions between the FlgE molecules must secure the stability while enabling the dynamic nature of the hook[13]. Previous studies on the hook of *Salmonella enterica* serovar Typhimurium (*S. enterica*) and of its protein, FlgEst, showed how some of the domains of FlgEst interact in the hook structure[6,14,15]. However, this structure lacks an important domain that could not be traced, limiting our understanding of the hook.

Flagella are found in both gram-positive and gram-negative bacteria. Although flagellar hooks appear identical at first sight, the diversity of flagellar hook proteins suggests that the hooks have diverged to specifically fit the motility requirements of each bacterium. The cells of the food-borne pathogen, *Campylobacter jejuni* (*C. jejuni*) are spiral-shaped and are able to move using unipolar or bipolar flagella, in comparison with rod-shaped *S. enterica* cells, which move using many peritrichous flagella over the cell surface[16,17]. Uniquely, the *C. jejuni* flagellar hook is also used to export virulence factors during colonization of the avian or human host[18,19]. Intriguingly, the *C. jejuni* hook protein, FlgEcj, has one of the longest amino acid sequences compared with other bacterial FlgE proteins[20]. Compared with FlgE from *S. enterica*, FlgEcj is about twice as large. We solved the structure of the hook of a *fliK* null mutant strain derived from *C. jejuni* strain 81116 (NCTC 11828) by electron cryo-microscopy, using single-particle averaging methods with image segment classification followed by systematic symmetry exploration.

## Results

**Structure determination.** The hook was purified from a mutant strain of *C. jejuni* 81116 in which the *fliK* gene encoding the FliK protein that regulates the hook length[8] has been deleted, thus producing a hook much longer than that of the wild-type strain. The sample was vitrified by plunge-freezing in liquid ethane and imaged at liquid nitrogen temperature under low dose conditions on a direct electron detector operating in movie mode on a Titan Krios cryo-TEM (transmission electron microscopy; FEI Company; Fig. 1a). Drift-corrected whole-frame averages of digitally extracted helical assemblies were processed with single-particle averaging methods using image segment classification followed by systematic symmetry exploration. Three dimensional (3D)-reconstruction and iterative refinement was applied using two independent data sets that were combined in the final reconstruction (see 'Methods' section for more details). The final combined reconstruction yielded a map at 3.5 Å resolution that was used to build the model of FlgE. We first built a model of FlgEcj of strain 81116 by structural homology with Swiss-Model[21], using the X-ray structure of a 79 kDa fragment of FlgE from *C. jejuni* strain NCTC 11168 that was solved by X-ray crystallography[22] as a template. The FlgE proteins from both strains of *C. jejuni* (81116 and NCTC 11168) have a sequence identity of 95%. However, FlgE of strain 81116 (NCBI Reference

Sequence: WP_012006803.1), presented here, is 851 amino acid residues long compared with 864 amino acids for FlgE of strain NCTC 11168 (NCBI Reference Sequence: WP_002882660.1). This homology model, which lacks about 90 and 40 amino acid residues in the N- and C- terminal chains, respectively, was first fitted as a rigid body into the 3.5 Å resolution map obtained by cryo-electron microscopy (Fig. 1b, Supplementary Fig. 1). The quality of the map allowed us to directly trace the missing parts of the initial model and to refine the complete structure, including side chains with real-space fitting methods.

**Structure of FlgE protein of *Campylobacter jejuni*.** The structure of the hook protein of *C. jejuni* shows five distinct domains: D0, D1, D2, D3 and D4 (Fig. 1c). Domain D0 consists of two bundled α-helices made by the N-terminus [Met1-Ser26] and the C-terminus [Leu815-Gln851]. These helices are about 44 Å and 51 Å long, respectively. Domain D0 also has long segments made by the N-terminal segment [Asn27-Thr91] and the C-terminal segment [Thr805-Asp814]. These segments connect the domain D0 to domain D1 and could not be traced in the previous structural studies of the hook due to a lack of resolution[14,15]. The C-terminal α-helix is directly linked to domain D1. The segment connecting the N-terminal helix to D1 first bends back and runs parallel to the helix, then bends again forming a hairpin that connects to D1. This N-terminal connecting segment forms an elongated 'L-shaped' structure made of a combination of two anti-parallel β-strands. We termed it the 'L-stretch'. The first part of the L-stretch, [Gly32-Ser45] and [Gly67-Gln81], is about 55 Å long and is roughly parallel to the N-terminal α-helix in D0. The second part, [Ser45-Gly67], is about 35 Å long and tilts slightly away from the two-helix bundle of D0. The sequence of domain D0 is well conserved throughout the bacterial world. Thus it is likely that the 3D structure of this domain is conserved.

The other domains, D1, D2, D3 and D4, are similar to those found in the crystal structure of the 79 kDa fragment of FlgE from *C. jejuni* strain NCTC 11168 (ref. 22). However, the relative position of these domains is different from their position in the crystal (Supplementary Fig. 2). The most significant difference is in domain D4 that is shorter in sequence by 14 amino acid residues compared with that of the crystallized strain. While domains D0, D1 and D2 are conserved within flagellated bacteria, domains D3 and D4 are more often found in FlgE proteins of Epsilonproteobacteria that have a *flgE* gene similar in length to that of *C. jejuni*, such as *Helicobacter*.

**Structure of the hook of *Campylobacter jejuni*.** The hook of *C. jejuni* is composed of 11 protofilaments that run along its length (Fig. 2a). When viewed along its axis, it can be described as a packing of concentric ring structures representing the different domains that make FlgEcj (Fig. 2b–f). The diameter of the hook is 280 Å, which is substantially wider than the hook of *S. enterica* that is 180 Å wide. Domain D0 makes the compact inner ring. It is almost entirely covered by domain D1, with the tip of the L-stretch protruding. Successive rings made by domains D2, D3 and D4 cover the surface with increasing radii, giving to the hook of *C. jejuni* its bulky helical tubular structure (Supplementary Movie 1). Although FlgEcj has two extra domains, D3 and D4, that do not exist in most other bacterial FlgE proteins, domains D0, D1 and D2 are structurally well conserved[22]. All of the molecular interactions in the hook involving these three domains can be generalized to most bacterial hooks. In this model, domains D1 and D2 have helical packing similar to that found in the hook of *S. enterica*[6,14,15], with a strong interaction in the six-start helical direction between the

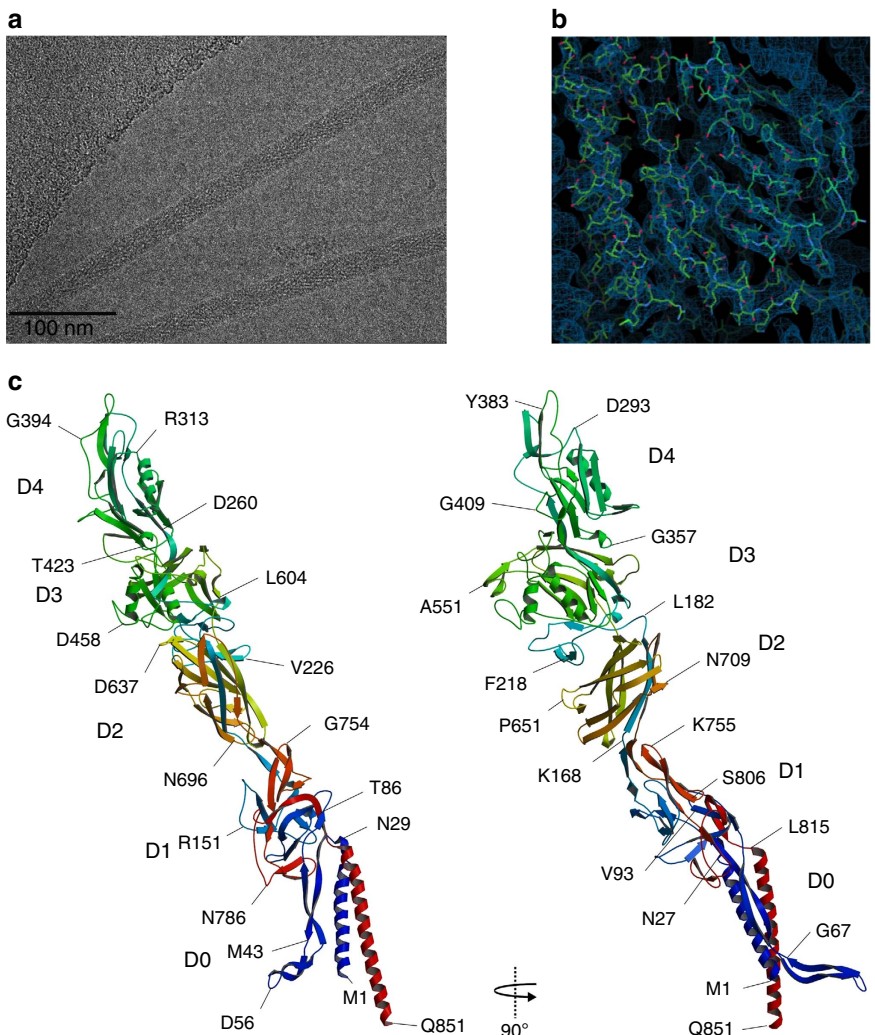

**Figure 1 | Complete structure of FlgE from *C. jejuni*.** (**a**) Cryo-electron micrograph of FlgEcj. (**b**) Cryo-EM density map superimposed with part of the structure of FlgEcj. (**c**) Two different views of the Cα backbone tracing of FlgEcj with its five domains, D0, D1, D2 D3 and D4. The chain is colour coded in a spectral ('rainbow') sequence, from blue at the N-terminus through red at the C-terminus. Figure prepared Figure prepared with Coot[36], MOLSCRIPT[44] and RASTER3D (ref. 45).

D2 domains of neighbouring protofilaments (Supplementary Fig. 3). However, the presence of domains D3 and D4 increase the number of interactions between the molecules of FlgE of *C. jejuni*, both within and between protofilaments. These interactions are specific to the hook of *C. jejuni*.

**The inner core in domain D0 and the central channel.** The inner core of the structure is made by D0, containing the two α-helices of the N-terminus and the C-terminus of FlgE, and the L-stretch. These helices are not parallel to each other but are close enough to interact in their respective central portions. During flagellar assembly, the export of substrates occurs through a central channel that runs along the helical axis of the flagellum[7]. In the hook, this central channel is made by the C-terminal α-helices from different protofilaments. These α-helices are closely packed and make the channel look like a continuous and smooth hollow structure with a diameter of 16 Å (Supplementary Fig. 4). Side chains of three amino acid residues are protruding into the channel: Gln830, Ser837 and Gln848. Residues Arg827, Ser833, Lys834, Asp840, Gln841, Gln844, Thr845, Gln851 are also lining the channel, making the inside channel surface hydrophilic. These residues are either

conserved in *S. enterica* or substituted with another polar amino acid residue (Supplementary Table 1). Because the C-terminal α-helices are organized like roofing tiles, only the bottom two-thirds of the α-helix makes the surface of the central channel.

In domain D0, the first part of the inner core is made by the C-terminal α-helix. This inner core is wrapped by the N-terminal helix (Supplementary Movie 2). These α-helices have different sets of interactions with other FlgE molecules (Fig. 6a,b). Along the '11-start' direction (within a protofilament) we have interactions between the end and the start of C-terminal α-helices (Fig. 3c). There are also interactions in the five-start and six-start directions, all exclusively between C-terminal α-helices (Fig. 3d,e). Between the N-terminal α-helices, only interactions in the five-start direction are formed (Fig. 3f). The only interactions between the N-terminal and the C-terminal α-helices are within the same molecule of FlgE.

**L-stretch has a dominant role in hook stabilization.** The L-shaped stretch (L-stretch) [Gly32-Gln81] in domain D0 of FlgE is the segment of the hook structure that has the most contacts between the different FlgE molecules making the hook. The L-stretch is engulfed in a pocket surrounded by six molecules of

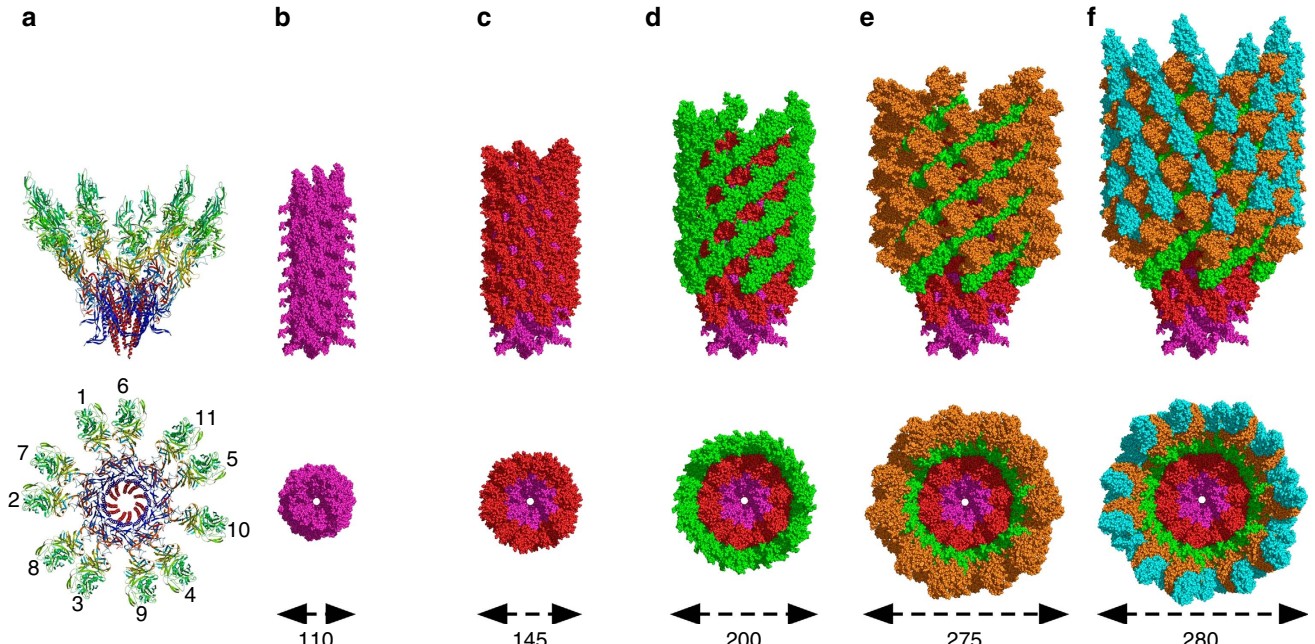

**Figure 2 | Domain distribution in the hook of *C. jejuni*.** (**a**) Lateral and top views of two turns of the hook of *C. jejuni* showing the eleven molecules that mark the start of the eleven protofilaments of the hook. (**b**–**f**) Lateral and top views of the hook of *C. jejuni* showing the organization and the packing of the different layers made by the domains of FlgE. Domain D0 in purple, domain D1 in red, domain D2 in green, domain D3 in brown and domain D4 in cyan. The image at left shows only D0. Additional domains are added in each subsequent image. The approximate diameters are indicated at the bottom. Figures were prepared with PyMOL (The PyMOL Molecular Graphic System, Schrödinger, LLC. http://www.pymol.org.).

FlgE that belong to four different protofilaments. Some of these interactions are between L-stretch segments in the 11-start and in the 5-start helical directions (Fig. 4). The other interactions are between the L-stretch and different segments located mostly in domain D1 of the other subunits (Fig. 5). In the minus five-start direction (Supplementary Fig. 3), most of the L-stretch has extensive interactions with the segments Asn90-Arg115 and Asn770-Ala785 (Fig. 5b), both located in domain D1. The tip of the L-stretch interacts with FlgE molecules in the minus 16-start and the minus 10-start directions (Fig. 5c). Some of these interactions are with FlgE molecules located two protofilaments away. Although the N- and C-terminal helices are very important for the polymerization of the hook[5], the complete 3D structure of the hook suggests that the L-stretch plays a dominant role in stabilizing the overall structure of the hook. These extensive contacts between the L-stretch domains of FlgE molecules might play a dual role by enabling, at the same time, the function of the hook as a universal joint. Sequence alignment of FlgE proteins from different bacteria shows the existence of the L-stretch domain in all species, although the sequence identity is low. All the interactions described above are expected to be present in diverse bacteria. Only four amino acid residues in the L-stretch are conserved: Thr30, Gly32, Lys34 and Phe40. Amino acid residues Phe33 and Met43 are occasionally replaced by similar residues Tyr or Leu, respectively (numbering is based on the sequence of *C. jejuni* FlgE).

**Sets of interactions from domains D3 and D4.** The two extra domains, D3 and D4, are only found in FlgE protein of *C. jejuni* and of related Epsilonproteobacteria with a relatively high sequence homology. These two domains introduce sets of additional interactions between the molecules that make the hook in these bacteria (Fig. 6). Within the same protofilament, in the 11-start direction, there are interactions involving domains

D3–D1, D3–D2, D4–D2 and D4–D3 between the lower molecule and the one just above it (Fig. 6a).

Between neighbouring protofilaments there are interactions in the 17-start helical direction between domains D4–D2 and D4–D3 (Fig. 6b,c). These interactions are made possible because of the length of FlgE protein of *C. jejuni*. They do not exist in the hook of *S. enterica* (Supplementary Fig. 5). There are also interactions in the six-start direction between domains, D3–D3, D4–D3 and D2–D2 (Fig. 6d).

Amino acid sequence variability in the central parts of FlgE proteins of *C. jejuni* strains was proposed to occur because of selection pressure during host invasion to generate variations in surface-exposed antigenic determinants[20]. The variable regions do indeed correspond to the surface-exposed region of *C. jejuni* hook domains D3 and D4. The variability is tolerated because these regions are not essential for intra-molecular contacts that organize the hook.

The new interactions within and between protofilaments could make the *C. jejuni* hook stiffer and stronger when rotating to transmit torque to the filament, while still allowing it to curve. The extra domains, D3 and D4, might slightly reduce the curvature of each protofilament, compared with the *S. enterica* hook.

**Domains D3 and D4 increase robustness of the hook.** The complete structure of the hook of *C. jejuni* has revealed extensive sets of inter-molecular interactions due to the existence of two extra domains: D3 and D4. Because most bacterial hook proteins do not have domains D3 and D4, we investigated the effects of deletion of these domains. A *C. jejuni* Δ*flgE*::Km^R null mutant strain (CB991) was made, and two derivative mutant strains from the Δ*flgE*::Km^R mutant strain were made: the first mutant strain (CB-A9) encodes the wild-type *flgE* gene integrated into the chromosome within the rRNA gene cluster; and

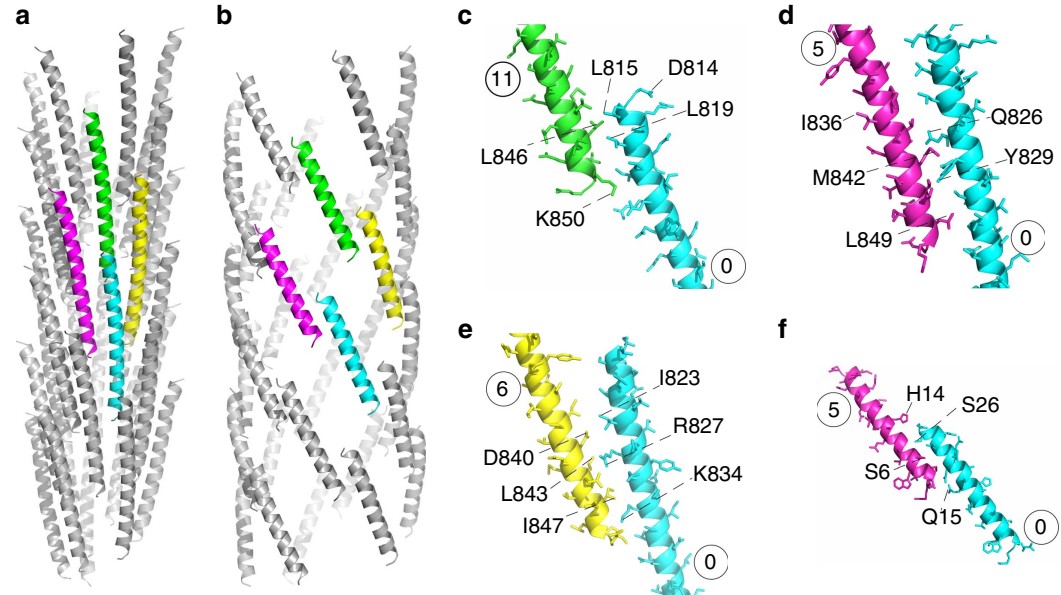

**Figure 3 | D0 interactions in the hook.** Inter-molecular interations between FlgE proteins in the central core between the C-terminal (**a**) and between the N-terminal (**b**) α-helices of domain D0 taken from 33 molecules. The coloured helices highlight these interactions in the 11-start (**c**), in the 5-start (**d**) and in the 6-start (**e**) direction for the C-terminal α-helices. The N-terminal α-helices only interact in the 5-start direction (**f**). The numbers indicate the order of appearance of each FlgE molecule during the building of the hook. Figure prepared with PyMOL.

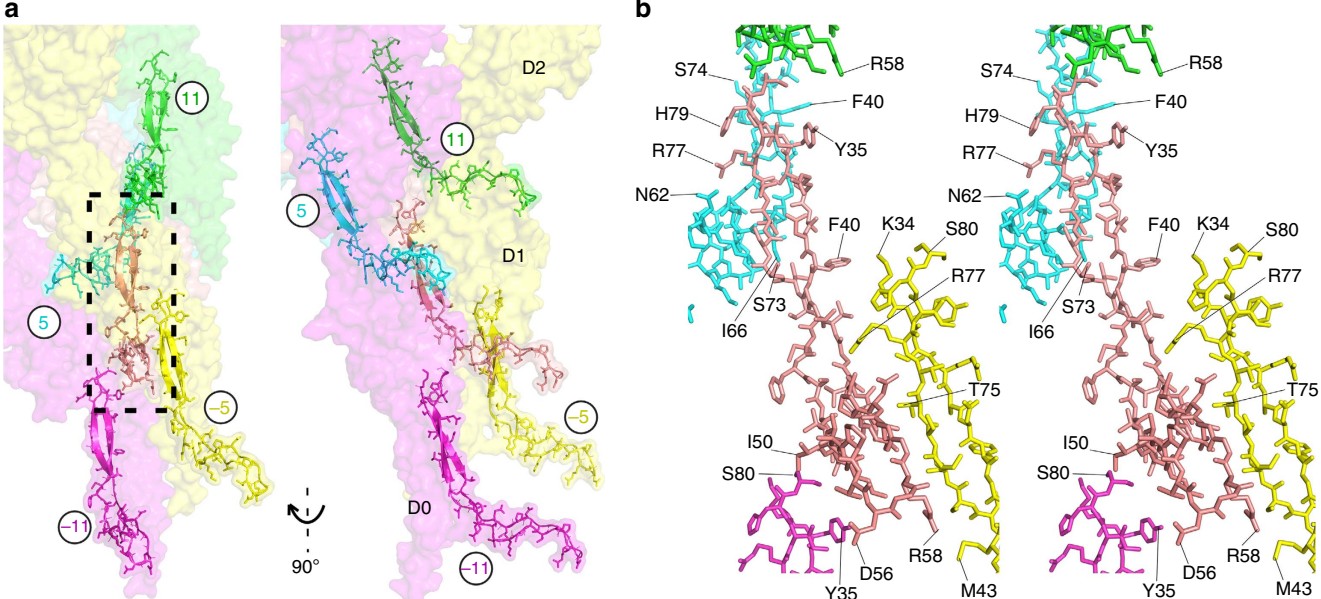

**Figure 4 | Interactions between L-stretch motifs.** The L-stretch interacts with other L-stretch motifs located in neighbouring protofilaments. (**a**) Overview of the position of the molecules involved in these interactions that are all restricted to domain D0. (**b**) A blown-up stereo view of the interactions between L-stretch motifs. These interactions are in the 5-start and in the 11-start directions, in the surrounding protofilaments and within the same protofilament, respectively. The circled numbers indicate the order of appearance of each FlgE molecule during the building of the hook (Supplementary Fig. 3). Figure prepared with PyMOL.

the second mutant strain (CB-A46) encodes a mutant *flgE* gene deleted for codons for the D3 and D4 domains integrated onto the chromosome within the rRNA gene cluster (Fig. 7). CB-A46 synthesizes a mutant FlgE protein with Ala235 fused to Gln603. CB-A9, which expressed the *flgE* gene from the alternative site on the chromosome was fully motile in Mueller-Hinton (MH) motility agar (Fig. 7a), and examination of negatively-stained cells using electron microscopy revealed that cells produced normal flagella (Fig. 7f). CB-A46, which

expressed *flgE* with codons for the D3 and D4 domains deleted was very poorly motile (Fig. 7a,b), and examination of the cells using electron microscopy revealed that some cells made flagella normally, but more than 90% of cells had short flagellar stubs at the base (Fig. 7g) and many broken flagella were seen. The breakage of the flagella seems likely to result from the absence of all extra sets of interactions within a protofilament and between protofilaments that have been described in this work.

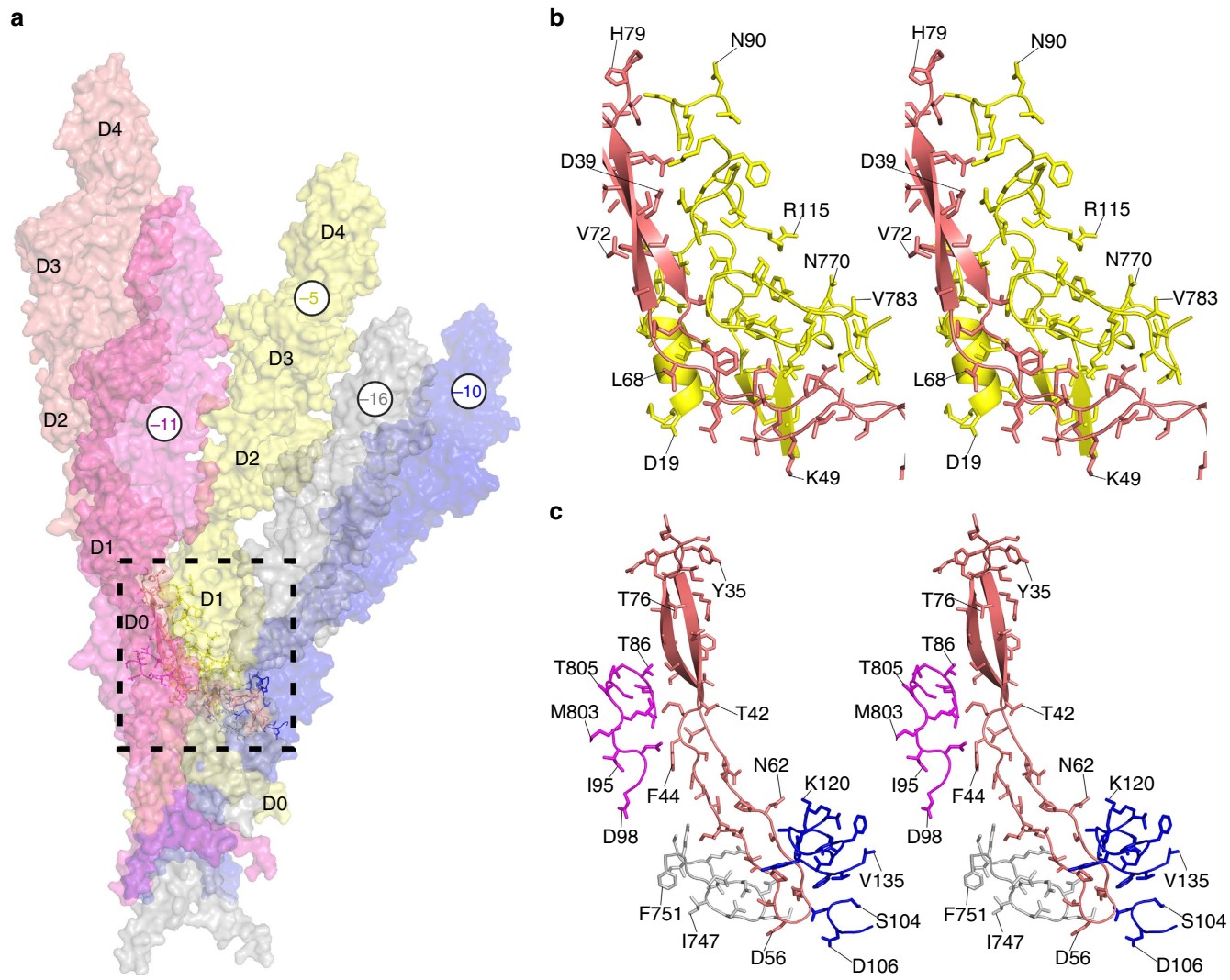

**Figure 5 | Global interactions of the ʟ-stretch.** The ʟ-stretch has extensive contacts with segments located in domain D1. (**a**) Overview of FlgE molecules that interact with the ʟ-stretch motif through segments of their D1 domain. (**b**) A blown-up stereo view showing part of domain D1 (in yellow) of the FlgE (in position −5) that enlaces the ʟ-stretch. (**c**) A blown-up stereo view showing segments of D1 domains of molecules at position −10 (blue) and −16 (grey), which interact with the tip of the ʟ-stretch, and at position −11 (magenta) that interacts with the middle part of the ʟ-stretch. The circled numbers indicate the order of appearance of each FlgE molecule during the building of the hook. Figure prepared with PyMOL.

## Discussion

The helical parameters of the hook from *C. jejuni*, 4.185 Å in rise and 64.34 degrees in rotation, are similar to those of the hooks of *S. enterica* and of *Caulobacter crescentus*[14,15,23,24]. The hook of *C. jejuni* has many sets of inter-molecular interactions. On the basis of the conserved structural domains among bacterial FlgE, some of these interactions are common to all bacterial hooks. However, the structure presented also shows interactions more specific to *C. jejuni* that will most likely have the effect of this hook having a larger radius of curvature, and being stiffer compared with a hook made from a FlgE protein with fewer domains. As it has been previously described the interactions between the molecules that make the hook are transient interactions where interacting amino acid residues will constantly change partners during the rotation of the hook[6].

Looking at the structure of the hook we notice that the ʟ-stretch domain of the FlgE molecules located at the base of the hook (at the proximal end) are not covered by other domains, unlike the ʟ-stretches located at the distal end. On the basis of the overall structure of the flagellum, we predict that the ʟ-stretch segments at the proximal end will interact with the distal rod

protein, FlgG. This will assure a strong set of interactions between the hook and the rod. A homology model of *C. jejuni* FlgG, built by using the D0 and D1 domains of FlgE, supports the idea of the interaction between FlgE and FlgG (Supplementary Fig. 6). The FlgG model has only two domains, D0 and D1, with a segment similar to the ʟ-stretch of FlgE as predicted from the primary sequence. The segment of FlgG as well as the D0 and D1 domains of FlgG could participate in making a tight connection to FlgE without the need for a junction protein.

Flagella, although macroscopically similar, have evolved features that will make them specially adapted to particular tasks. The intestinal jejunum is a viscous environment where *C. jejuni* is adapted for swimming[25,26]. The results shown here tend to support the idea that additional strengthening of the hook in *C. jejuni* is necessary to enable motility in this viscous environment.

The axial part of the bacterial flagellum is a continuous, helical structure that starts with the rod proteins, FlgB, FlgC, FlgF and FlgG, continues through the hook, the hook-associated proteins HAP1 and HAP3, the filament, and ends with HAP2, which caps the distal end of the filament[7]. This organization is consistent

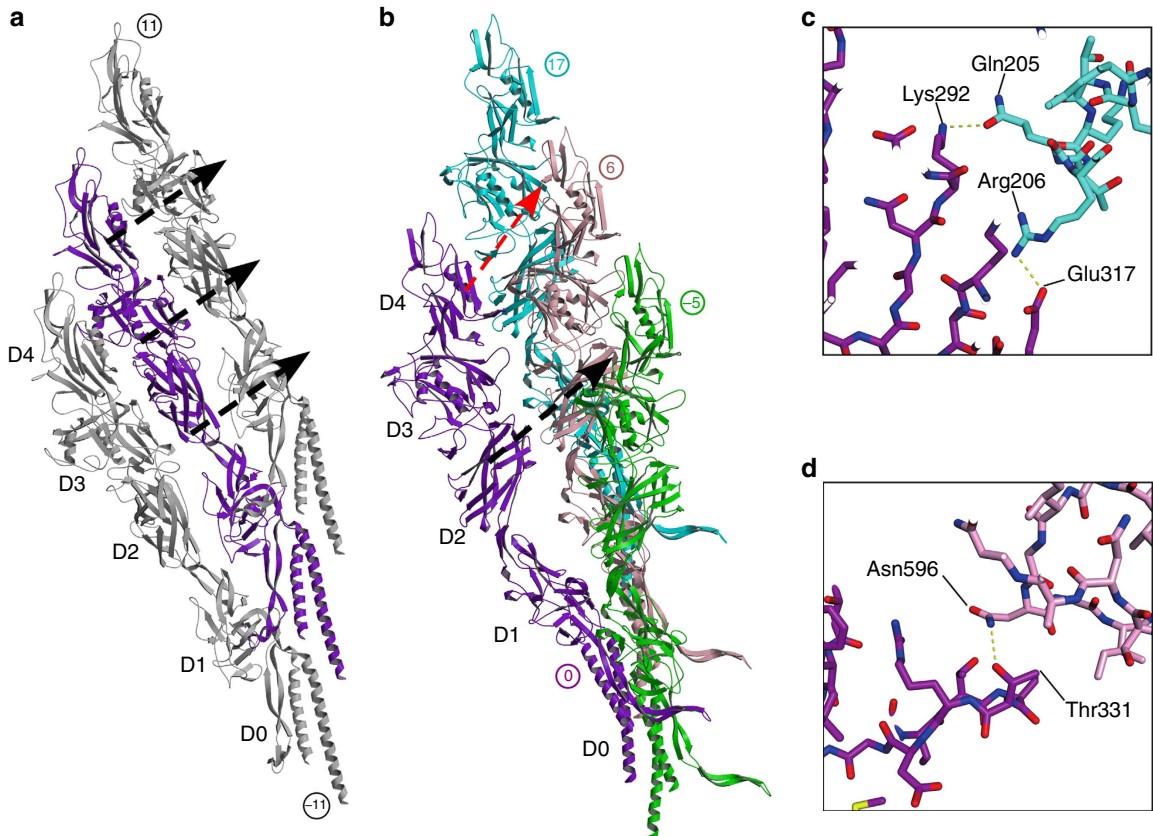

**Figure 6 | Interactions within and between protofilaments.** Two protofilaments taken on each side of the hook and showing the interactions in the 11-start direction between domains D2–D1, D3–D2 and D4–D3 (**a**). Four molecules of FlgE taken from two neighbouring protofilaments with the interactions in the 6-start (black dashed arrow) and in the 17-start (red dashed arrow) (**b**). The numbers indicate the order of appearance of each FlgE molecule during the building of the hook. Figure prepared with MOLSCRIPT[44] and RASTER3D (ref. 45). Closed-up views of side-chain interactions in the 17-start direction between domain D4–D3 (**c**) and the 6-start interaction between D2–D2 (**d**). Figures are prepared using the same colours used in **b** with PyMOL.

with the hook and the rod both having an 11-protofilament structure. FlgG forms the distal part of the rod, which connects the hook to the motor[7]. However, with the hook of *C. jejuni* consisting of an assembly made of 11 protofilaments and its filament having 7 protofilaments[27], *C. jejuni* and most likely related organisms as well, could be the only known bacteria with the hook and the filament having a different quaternary structure. This will result in a symmetry mismatch, a structural conundrum. This highlights the roles of the hook-associated protein 1 and 3 (HAP1 and HAP3) that form the junction between the hook and the filament. HAP1 and HAP3 function is to connect segments of the flagellum with different 3D structure and mechanical characteristics[6,28,29]. In the case of *Campylobacter*, HAP1 and HAP3 would play the extra role of linking the hook and the filament despite their different quaternary symmetry. High-resolution structures of HAP1, HAP3 and the filament will be necessary to understand the connectivity between the hook and the filament.

## Methods

**Bacterial strains and plasmids.** All bacterial strains and plasmids used in this study are described in Supplementary Table 2. Oligonucleotides used in the strain and plasmid constructions are listed in Supplementary Table 3. *C. jejuni* 81116 and mutant-strain derivatives were cultured on MH (Difco, Detroit, MI, USA) or Campy blood-free selective medium (CCDA; Oxoid, Basingstoke, Hampshire, UK) plate media and incubated at 37 °C under microaerophilic conditions maintained by using AnaeroPack (Mitsubishi Gas Chemical, Tokyo, Japan). For *Campylobacter* antibiotics were used at the following concentrations, where appropriate: trimethoprim, 5 μg ml$^{-1}$; vancomycin, 10 μg ml$^{-1}$; chloramphenicol, 17.5 μg ml$^{-1}$; kanamycin, 50 μg ml$^{-1}$; and apramycin, 60 μg ml$^{-1}$. *E. coli* NovaBlue

was grown in Luria-Bertani (LB) agar or broth[30]. For *E. coli*, ampicillin, 50 μg ml$^{-1}$ was added to media where appropriate.

**Preparation of polyhook for cryo-electron microscopy.** A *fliK*-deficient strain of *C. jejuni* 81116 was cultured on MH or CCDA plate media and incubated at 37 °C under microaerophilic conditions. After 48–72 h cultivation, the polyhooks were isolated by the previously reported methods with modifications[15,31]. In brief, cells were removed from the plates and washed with phosphate-buffered saline solution. After the cells were spun down, the cells were suspended in a Tris–HCl (pH 8.0; Sigma-Aldrich, St Louis, MO, USA) buffer containing sucrose (Wako, Osaka, Japan) supplemented with lysozyme and magnesium chloride for spheroplast formation. This step was followed by adding Triton X-100 (Sigma-Aldrich, St Louis, MO, USA) for cell lysis. Polyhooks were released from cell debris by alkaline treatment (pH 11.0) and pelleted by ultracentrifugation. Finally, polyhooks were suspended in a Tris–HCl (pH 8.0) buffer containing EDTA (Nippon Gene, Toyama, Japan) and Triton X-100.

**Specimen preparation for cryo-EM.** Purified polyhook solution of 4 μl were applied to a holey carbon grid (C-Flat CF-1.2/1.3, Protochips Inc) previously treated for 20 s with a hydrogen/oxygen plasma (80/20%) in a Gatan Solarus plasma cleaner and plunged into liquid ethane (cooled to − 178 °C) after blotting from both sides with filter paper (Whatman #40, 95% humidity, 4 °C, 25 s blot time) using a Gatan CP3 vitrification system. Grids were mounted submerged under liquid nitrogen in autogrid cartridges (FEI Company) and transferred to a Titan Krios cryo-TEM (FEI Company) in their specimen cassette.

**Imaging.** The Titan Krios was operated in TEM mode at 300 kV after standard alignments including coma-free alignment. The specimen was kept at liquid nitrogen temperature. Data acquisition was performed with the *Leginon* system. Briefly, areas of thin ice were identified after mapping the grid at low magnification to produce a grid atlas. Suitable meshes were marked and their hole pattern indexed by pattern recognition. An additional hole-imaging step allowed manual targeting of individual helices, which were added to the imaging queue. Off-axis

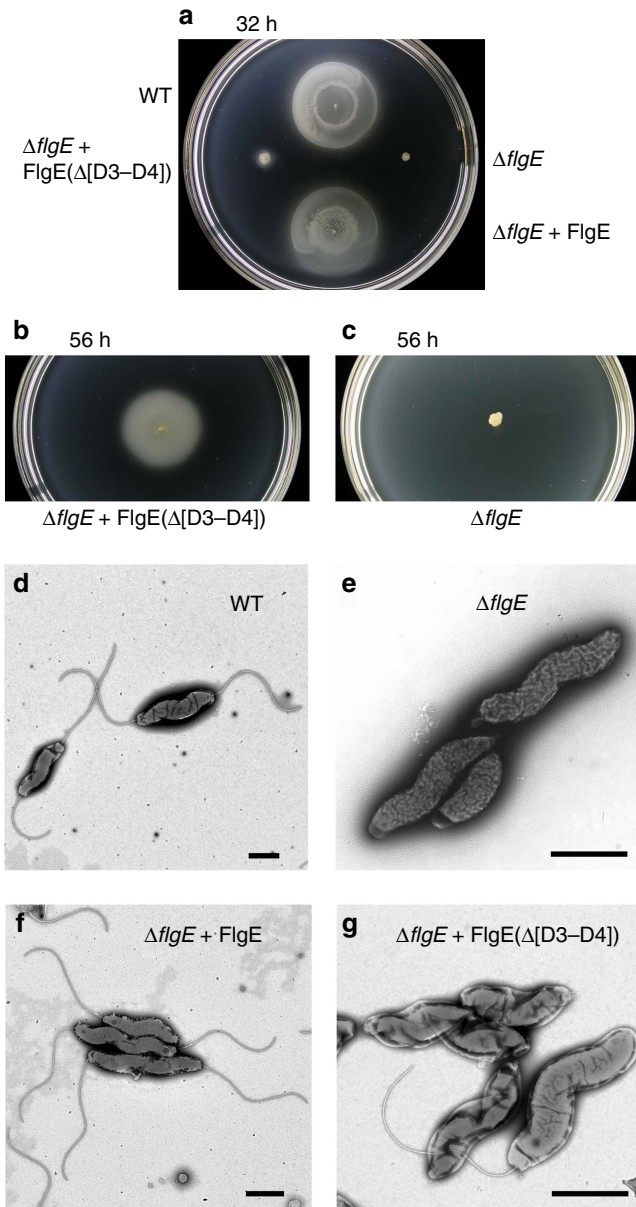

**Figure 7 | Motility and flagella of *C. jejuni* expressing FlgE with domains D3 and D4 deleted.** (**a**) Motility phenotype of *C. jejuni* strains stab inoculated into motility agar after 32 h at 37 °C. The following strains were examined: the wild-type strain; the Δ*flgE*::Km$^R$ mutant strain; the Δ*flgE*::Km$^R$ + FlgE mutant strain; and the Δ*flgE*::Km$^R$ + FlgE(Δ[D3–D4]) mutant strain, which encodes FlgE deleted for the D3 and D4 domains. (**b**) Motility phenotype of the Δ*flgE*::Km$^R$ + FlgE(Δ[D3–D4]) strain and (**c**) the Δ*flgE*::Km$^R$ strain inoculated into motility agar both after 56 h at 37 °C. The wild-type strain and the Δ*flgE*::Km$^R$ + FlgE mutant strain covered the plate after 56 h at 37 °C and are not shown. (**d**) Negatively-stained wild-type cells examined by transmission electron microscopy with bipolar flagella. (**e**) Cells of the Δ*flgE*::Km$^R$ strain without flagella. (**f**) Cells of the Δ*flgE*::Km$^R$ + FlgE strain with bipolar flagella. (**g**) Cells of the Δ*flgE*::Km$^R$ + FlgE(Δ[D3–D4]) strain with one long flagellum or short flagellar stubs at the cell poles. At least 30 cells were examined for each strain and representative cells are shown. Scale bars, 1 μm.

auto-focusing (defocus range − 0.5 to − 2.5 μm) preceded every third data image. We used nano probe mode with parallel illumination (field-emission gun extraction 3,700 V, spot 5), beam spread to ∼1.5 μm diameter on the grid to illuminate the surrounding carbon with the goal to reduce charging effects. Image movies were recorded on a Falcon-II direct electron detector at a dose rate of 50 electrons

per pixel s$^{−1}$, for 1.5 s using raw frame capture at 18 fps at 79,000 × nominal magnification ( × 105,000 calibrated magnification, 1.08 Å per pixel and dose rate of 30 electrons Å$^{−2}$ s$^{−1}$ at the specimen level).

**Image processing.** Whole-frame drift correction was applied using the GPU-accelerated programme by Li *et al.*[32]. Initially, stacks of the first 13 aligned movie frames were summed and averaged (total dose of 35 electrons Å$^{−2}$). Aligned averaged images were manually selected for their contrast transfer function (CTF) properties and appearance of layer lines in their power spectra during estimation of CTF parameters (*CTFFIND3, CTFTILT*)[33]. A total of 1,166 helices were manually indicated on 434 selected digital micrographs using *E2HELIXBOXER* (ref. 34). All subsequent operations were performed with *SPRING* helical image processing package[35]. The selection of helices was narrowed to 550 individual helices with clearly visible layer lines. Images were convoluted with their fitted CTF and helices segmented at an interval of 25 Å, box size 700 pixel squared (total number of segments: 28,238). The stack of segmented helices was submitted to two-dimensional (2D) classification. A class with strong layer line amplitudes and crisp details was selected (Supplementary Fig. 7) for symmetry exploration. The helical operator was determined with a real-space search algorithm implemented in the *Segclassreconstruct* programme of SPRING, which calculates the correlation between a 2D class average image and the back projection from its symmetrized 3D reconstruction. On the basis of the helical parameters published for the hook of *S. enterica*[15], a range of combinations covering a rise, between 4.0 and 5.5 Å, and rotation between 63.0 and 67.0 degree, was systematically probed with this programme at increments of 0.1 Å for the rise and 0.1 degree for the rotation, respectively, using a 2 × binned class average reference image on the OIST high-performance computer cluster. Helical operators from solutions with the highest correlation were subsequently used in iterative independent 'gold standard' orientation refinement of the complete segmented particle data set with *SEGREFINE3D* (ref. 35) starting with 8 × binned data and leading to a reconstruction at 3.7 Å resolution (FSC = 0.14) with unbinned data. The helical parameters, which gave the highest resolution, was locally optimized by fine grained variation in 0.01 and 0.005 intervals for the rise and the rotation, respectively, followed by additional rounds of refinement with unbinned data. The refinement was repeated while including only the first five frames (total dose of 13 electrons Å$^{−2}$), yielding 3.5 Å resolution for a soft-masked reconstruction (Supplementary Fig. 7) with visibly more detail in the appearance of side chains as compared with a reconstruction including all frames. The final helical operator for the *C. jejuni* polyhook, rise 4.185 Å, rotation 64.34 degrees, was close to values reported for the hook of *S. enterica*[15]. These values depend on the pixel size, which has been calibrated to 1.08 Å per pixel at the specimen level with a Au/Pd replica grating (Ted Pella Nr.607) by measuring the diffraction ring in the power spectrum of the image.

**Modelling building and refinement of the *Campylobacter* hook.** The domains D1 through D4 of the hook protein of *C. jejuni* strain 81116 were built by structural homology with Swiss-Model[21], using the X-ray structure of a 79 kDa fragment of FlgE from *C. jejuni* strain NCTC 11168 that was solved by X-ray crystallography as a template (PDB-ID: 5AZ4). The FlgE proteins from both strains of *C. jejuni* (81116 and NCTC 11168) have a sequence identity of 95%. However, FlgE of strain 81116 (NCBI Reference Sequence: WP_012006803.1), presented here, is 851 amino acid residues long compared with 864 amino acids for FlgE of strain NCTC 11168 (NCBI Reference Sequence: WP_002882660.1). This homology model, which lacks 92 and 46 amino acid residues in the N- and C- terminal chains, respectively, was first docked as a rigid body into the 3.5 Å resolution map obtained by cryo-electron microscopy. The quality of the map allowed us to directly trace the missing parts of the initial model and to build the complete atomic model, including side chains with real-space fitting methods. The missing parts, D0 domain, which contains the N-terminal 92 residues and the C-terminal 46 residues that are missing in the X-ray structure and part of the D4 domain, were modeled manually using *Coot*[36]. During manual model building, stereochemistry of the peptide bonds was monitored with Ramachandran plot[37]. After all 851 residues of the hook protein of *C. jejuni* 81116 was completed, the structure was finally refined at 3.5 Å resolution with *phenix.real_space_refine*[38]. Ramachandran plot of the final structure shows 88.9 and 8.8% of residues are modeled at preferred and allowed regions, respectively. No rotamer and C$^β$ deviation outliers are found in the final structure. A map correlation coefficient around the model atoms was 0.764. *MolProbity*[39] evaluated all-atom clashscore of the final structure as 15.78 (Supplementary Table 4). The final structure was deposited in Protein Data Bank (PDB-ID: 5JXL).

**A suicide vector for knockout of the *flgE* gene.** The genome of *C. jejuni* 81116 (NCBI Reference Sequence: NC_009839.1) encodes a *flgE* gene (locus tag: C8J_RS08415) for the FlgE protein, which builds the hook described in this report. A suicide vector, pCB963, was made, so that the *flgE* gene could be deleted from the chromosome and replaced with a kanamycin-resistance gene. First, a plasmid was made from pUC19, called pCB951, which carries the *flgE* gene region. To make pCB951, a 3,849-bp DNA segment was amplified by PCR using *C. jejuni* 81116 genomic DNA as template with primers Fd-pUC19-CamflgEr and Rv-pUC19-CamflgEr. This encodes the 2,559-bp *flgE* gene, and 668-bp of flanking DNA

upstream and 622-bp of flanking DNA downstream of the *flgE* gene. Using a Gibson assembly cloning kit from New England Biolabs (USA) this DNA fragment was fused with pUC19 plasmid DNA that had been digested with Hind III and Kpn I to produce plasmid pCB951.

Plasmid pCB963 was made from pCB951 and it carries a kanamycin-resistance gene in place of the *flgE* gene, so that expression of the kanamycin-resistance gene is under the control of the *flgE* gene promoter. Using plasmid pCB951 as template, the 4,024-bp plasmid backbone (without the *flgE* gene) was amplified by PCR with primers Fd-pCB951-Linear and Rv-pCB951-Linear. The 795-bp kanamycin-resistance gene was amplified by PCR using plasmid pJMK30 as template with primers Fd-flgE-Km-swap and Rv-flgE-Km-swap. (Plasmid pJMK30 was generously provided by Julian M. Ketley, University of Leicester, UK.) The PCR product of the plasmid backbone of pCB951 was fused to the kanamycin-resistance gene PCR product using a Gibson assembly cloning kit to produce plasmid pCB963. Plasmid pCB963 encodes 668-bp DNA upstream, and 703-bp DNA downstream, of the kanamycin-resistance gene, from the *flgE* gene region. Standard molecular biology procedures were followed in the plasmid constructions[30].

**Suicide vectors for integration of genes onto the chromosome.** To complement *Campylobacter* mutant strains bearing chromosomal gene deletions, integration of the gene of interest within the ribosome cluster has been successfully used[40,41]. Plasmid pRRA was generously provided by Erin C. Gaynor (University of British Columbia, Canada), which can be used for gene integration within the ribosome cluster and carries a gene for apramycin-resistance[41]. However, the DNA sequence of pRRA encodes a fragment of the ribosome clusters from *C. jejuni* NCTC 11168 and there are many nucleotide substitutions in comparison with the DNA sequence of the three ribosome clusters of *C. jejuni* 81116. A new rRNA cluster integration vector was made, pCB952, which is pUC19 carrying '16S rRNA-Apr^R-tRNAala-tRNAile-28S rRNA' genes (16S rRNA and 28S rRNA genes are truncated at the 5′ and 3′ ends, respectively). Plasmid pCB952 is similar to pRRA, but encodes RNA genes from strain 81116.

Plasmid pCB950 was first made, which is pUC19 carrying genes, from the ribosome cluster only. Using *C. jejuni* 81116 genomic DNA as template with primers Fd-pUC19-81116-RNA and Rv-pUC19-81116-RNA, 1,899-bp DNA encoding the rRNA genes was amplified by PCR. This DNA fragment was fused with pUC19 plasmid DNA that had been digested with Hind III and Kpn I using a Gibson assembly cloning kit to produce plasmid pCB950. Plasmid pCB952 also carries a gene for apramycin resistance and it was made from plasmid pCB950. Using plasmid pCB950 as template, the 4,556-bp plasmid backbone was amplified by PCR with primers Fd-pCB950-Linear and Rv-pCB950-Linear. The 907-bp apramycin-resistance cassette was amplified by PCR using plasmid pRRA as template with primers Fd-Apr-cassette and Rv-Apr-cassette. The PCR product of the plasmid backbone of pCB950 was fused to the apramycin-resistance cassette PCR product using a Gibson assembly cloning kit to produce plasmid pCB952. Plasmid pCB952 encodes 805-bp DNA upstream, and 1,098-bp DNA downstream, of the apramycin cassette, from the ribosome cluster.

A suicide vector, pCB956, was made from pCB952 to integrate the *flgE* gene within the ribosome cluster under the control of the *flgE* gene promoter. Using plasmid pCB952 as template, the 5,463-bp plasmid backbone was amplified by PCR with primers Fd-pCB952-Linear and Rv-pCB952-Linear. A 2,786-bp DNA fragment encoding the *flgE* gene was amplified by PCR using *C. jejuni* 81116 genomic DNA as template with primers Fd-flgE-Int and Rv-flgE-Int. The PCR product of the plasmid backbone of pCB952 was fused to the *flgE* gene-encoding PCR product using a Gibson assembly cloning kit to produce plasmid pCB956.

A suicide vector, pCB966, was made to integrate a mutant *flgE* gene without codons for the D3 and D4 domains onto the chromosome, within the rRNA cluster. Using plasmid pCB956 as template and primers Fd-CamflgE(1810–1832) and Rv-CamflgE(691–708) the codons for the D3 and D4 domains were deleted using a Q5 site-directed mutagenesis kit (New England Biolabs, USA).

**Construction of *Campylobacter* mutant strains.** To make the desired mutant strain of *C. jejuni*, the parent strain was transformed with plasmid DNA by electroporation as previously described[42]. Because the plasmids were derived from pUC19 they do not encode a replication origin specific for *C. jejuni* and are therefore unable to replicate in *C. jejuni* cells. Instead they acted as suicide vectors and since they carried DNA sequence homologous to that on the chromosome, homologous recombination occurred between the plasmid and chromosome. Selection of mutant-strain derivatives after electroporation, bearing the antibiotic resistance gene integrated onto the chromosome was performed on MH agar containing the appropriate antibiotics. Colony-forming units of the mutant derivatives were purified. It was confirmed using PCR and primers annealing to chromosomal DNA at sites upstream and downstream of the region of integration that double-crossover homologous recombination had occurred upstream and downstream of the antibiotic resistance gene at the correct location on the chromosome. It was verified that the DNA sequences of the PCR products were correct by chain-termination dideoxynucleotide sequencing using a BigDye Terminator v3.1 cycle sequencing kit (Thermo Fisher Scientific, USA). *C. jejuni* 81116 was electroporated with plasmid pCB963 to make strain CB991. Strain CB991 was electroporated with plasmids pCB956 or pCB966 to produce strains CB-A9 or CB-A46, respectively.

**Motility assays.** Motility of *C. jejuni* strains was examined similarly to a previously described method[43]. The strains were streaked on MH agar, containing antibiotics as required, and grown at 37 °C under microaerophilic conditions for 48 h. Each strain was suspended from the plate in MH broth to an $OD_{600}$ nm of 1.0 and 1 μl was stab inoculated into MH motility medium containing 0.4% (w/v) agar. Motility phenotypes were examined after incubation of strains at 37 °C under microaerophilic conditions for the desired time. Experiments were repeated at least six times for each strain.

**Transmission electron microscopy of whole cells of *C. jejuni*.** The presence of flagella on *C. jejuni* 81116 and derivative mutant strains was examined similarly to a previously described method[19]. Briefly, the strains were grown on MH agar, containing appropriate antibiotics, at 37 °C under microaerophilic conditions for 48 h. A 100 μl aliquot of water was used to suspend the cells, and the cells were gently scraped from the surface of the plate using a sterile plastic inoculating loop. The cells were pelleted by gentle centrifugation (1,000g for 5 min) and suspended in 100 μl water. Cell suspension was spotted on Mextaform HF-34 200-mesh carbon-coated copper grids and the cells were directly stained with 1% phosphotungstic acid at pH 7.0 for 2 min. The excess liquid was removed by blotting with Whatman filter paper. Grids were examined using a JEM-1230R transmission electron microscope (JEOL, Ltd., Japan) at 100 kV. At least 30 cells were examined for each strain. Images of whole cells were photographed at between × 2,500 and × 10,000 magnification.

**Data availability.** Atomic coordinates have been deposited in the Protein Data Bank under accession code 5JXL.

The map of the *Campylobacter jejuni* hook has been deposited in the EM Data Bank under the accession number EMD-8179.

Interactive 3D views of the structure reported here: http://proteopedia.org/w/Samatey/4

All other relevant data is available within the contents of this manuscript, its Supplementary Files, or from the corresponding author on reasonable request.

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

## Acknowledgements

We acknowledge the members of the OIST Imaging and Materials Analysis Section for their continuous help in the characterization of our samples. We are grateful to Marc M. S. M. Wösten, from the University of Utrecht, The Netherlands, for providing us with the fliK mutant strain of C. jejuni. We would like to thank all the members of the Trans-Membrane Trafficking Unit for the useful discussions and for their help in preparing some of the samples. We are grateful to Keiichi Namba, from Osaka University, Japan, for his continuous support. We would like to thank Eric Martz and Steven D. Aird for critical reading the manuscript. Direct funding provided by OIST supported this work.

## Author contributions

F.A.S. conceived the project with the help of C.S.B. and H.M. H.M. prepared the sample. M.W. collected and processed the cryo-EM data. H.M. built and refined the model with the help of Y.-H.Y. and F.A.S. C.S.B. designed and made the experiments for the functional assays. All authors took part in the interpretation of the results. F.A.S. was responsible for the overall project strategy and management and wrote the manuscript with inputs from co-authors.

## Additional information

**Competing financial interests:** The authors declare no competing financial interests.

