## [Peer Review File · Nature Communications]

REVIEWERS' COMMENTS:

Reviewer #1 (Remarks to the Author):

Bacterial flagellar hook functions as the universal joint that transmits the torque from the flagellar motor to the filament. In this manuscript, Matsunami et al. reported the first complete atomic structure of a bacterial hook by using cryo-electron microscopy. The hook structure from *Campylobacter jejuni* not only uncovers the conserved core domains (D0, D1, and D2) of the hook protein FlgE for the first time, but also reveals the unique features (extra domains D3 and D4) in *C. jejuni* and related bacteria. The authors also provide evidence that the presence of the extra domains (D3 and D4) and their inter-molecular interactions are important for the flagellar assembly and rotation in *C. jejuni*. The manuscript is well organized and well written. The structures and the implication should be greatly welcomed in the fields of Microbiology and Structural Biology.

Specific concerns:

1. The atomic structure of the hook structure determined by cryo-EM and advanced helical reconstruction is amazing. It will be critical to describe the detailed procedure in determining the helical parameter of 4.18Å rise and 64.3 degree angular interval. Is there any difference between the helical parameter of the *C. jejuni* hook and that of the *Salmonella* hook?
2. Do all the hook segments share similar helical symmetry?
3. Figure 1 Legend: There's a typo and "figure prepared" is printed twice.
4. Figure 4: This figure is very confusing since there is no overview image like the one in A showing the L-stretch region with the relevant labelled D1 domains. Furthermore it is confusing to have the 0 l-stretch region go from orange to green, while yellow and magenta colors were reused to label different copies of FlgE than the copies they represent in A.
5. Figure 6: There is a noticeable motility in the D3-D4 mutant after incubation at 37C for 56 hours. Why does the motility of the D3-D4 mutant restore after additional 24-hour incubation?
6. Page 27; line 513: What do they mean by "Image CTF was re-determined at the new magnification?" wasn't CTF fit at bin1 initially? What new magnification? Was the magnification refined as well in SPRING and found to be so deviated as to require this?

Reviewer #2 (Remarks to the Author):

This is an excellent study that utilizes cryoelectron microscopy to determine the structure of the flagellar hook of *Campylobacter jejuni* at 3.5 angstroms. The results, which are described very clearly, demonstrate remarkable interactions of adjacent protofilaments that provide strength to this structure. *C. jejuni* must be motile in intestinal mucus in order to survive in the gut. The hook protein of *C. jejuni* was known to be larger than those of other bacteria. The extra domains of *C. jejuni* hook further enhance interactions with other protofilaments and the strength of the hook. Interestingly, the *C. jejuni* hook has an assembly of 11 protofilaments, while the flagella have an assembly of 7 protofilaments, which is the first example of hook and flagella with different quaternary structures.

This study is of major importance and of general interest to microbiologists and biochemists interested in protein structure. My comments are all minor:

1. line 44: "Dynamic" should be "dynamism" or "dynamic nature"
2. line 89-90: "These segments connect the domain DO to DI and were missing from all previous structural studies of the hook." The cited references are for Salmonella, so does this mean that these residues are not found in Salmonella? Please clarify
3. Fig 6 and corresponding text: line 203—do you mean Fig. 6f? line 205—do you mean Fig. 6g? Fig 6 a and b are confusing. The difference in motility of $\Delta\text{flgE}::\text{Km} + \text{flgE}(\Delta\text{D3-D4})$ on the two plates is due to the length of incubation (32 vs 56 h). Where is the control of wildtype at 56h? I presume that the motility is larger. What is the significance of these differences?
4. Line 392: "Figure prepared Figure prepared with Coot" needs correction.
5. Fig. 5: there is no legend for 5c and 5d.

Point-by-Point Response to Reviewers

We would like to thank the reviewers for their comments and, mostly, for helping us improve our manuscript.

Reviewers's comments are written below in blue.

Reviewer #1 (Remarks to the Author):

1. The atomic structure of the hook structure determined by cryo-EM and advanced helical reconstruction is amazing. It will be critical to describe the detailed procedure in determining the helical parameter of 4.18Å rise and 64.3 degree angular interval. Is there any difference between the helical parameter of the *C. jejuni* hook and that of the *Salmonella* hook?

The procedure used to calculate the helical parameters has been detailed in the “Material & Methods” section (Page 23, Line 516 ~ 537).

“The helical operator was determined with a real-space search algorithm implemented in the *Segclassreconstruct* program of SPRING, which calculates the correlation between a 2D class average image and the back projection from its symmetrized 3D reconstruction.”

The helical parameters for *Campylobacter* and for *Salmonella* hook are similar but different:
The helical parameters of the hooks from *Campylobacter* and *Salmonella* are very similar. This has been mentioned in the discussion, page 10 Line 208.

2. Do all the hook segments share similar helical symmetry?

All bacterial hooks that have been studied up to now (including this study) have similar helical parameters. The structure hooks from *Caulobacter crescentus*, *Salmonella enterica* and *Campylobacter jejuni* have been studied at different resolutions.

Caulobacter crescentus: rise 4.19 Å, rotation 64.38 degrees

Salmonella enterica: rise 4.12 Å, rotation 64.78 degrees

Campylobacter jejuni: rise 4.19 Å, rotation 64.35 degrees (current study)

3. Figure 1 Legend: There's a typo and "figure prepared" is printed twice.

The legend of "Fig. 1" has been corrected.

4. Figure 4: This figure is very confusing since there is no overview image like the one in A showing the L-stretch region with the relevant labelled D1 domains. Furthermore it is confusing to have the 0 l-stretch region go from orange to green, while yellow and magenta colors were reused to label different copies of FlgE than the copies they represent in A.

The "Figure 4" has been replaced by two figures: "Figure 4" and "Figure 5". Both figures give more details on the interactions of the L-stretch.

5. Figure 6: There is a noticeable motility in the D3-D4 mutant after incubation at 37C for 56 hours. Why does the motility of the D3-D4 mutant restore after additional 24-hour incubation?

There is no motility restoration. The "D3-D4" mutant strain (FlgE(Δ [D3-D4])) is motile but barely noticeable after an incubation for 32 hours at 37°C. For this reason, we made a separate swarming plate that ran for 56 hours in "Fig. 6b". The strain " Δ flgE::Km^R" was also left for 56 hours in "Fig. 6c" as a negative control for comparison.

We slightly changed the "Figure 6" and its legend to make this point clear to the readers.

6. Page 27; line 513: What do they mean by "Image CTF was re-determined at the new magnification?" wasn't CTF fit at bin1 initially? What new magnification? Was the magnification refined as well in SPRING and found to be so deviated as to require this?

During attempts to improve the resolution further, a magnification refinement against fitted long alpha helices in the D0 domain model was carried out. The CTF was re-calculated for the resulting slightly changed pixel size – particle segments were deconvoluted and a new reconstruction with two additional rounds of refinement starting from the latest alignment parameters was performed. However, the resolution of the final map did not improve significantly and this strategy was subsequently abandoned. The part of the manuscript referring to this magnification refinement had erroneously found its way into our manuscript. Thank you for spotting this inconsistency. This part of the sentence has now been removed.

Reviewer #2 (Remarks to the Author):

1. line 44: “Dynamic” should be “dynamism” or “dynamic nature”

This has been changed to “*dynamic nature*” on Page 3, Line 43

2. line 89-90: “These segments connect the domain DO to DI and were missing from all previous structural studies of the hook.” The cited references are for Salmonella, so does this mean that these residues are not found in Salmonella? Please clarify

The corresponding residues exist in the sequence of Salmonella but could not be traced in the structure of the hook of Salmonella because the resolution of the map obtained by cryo-electron microscopy was not high enough. The resolution was about 7 Å (Fujii et al., 2009).

This part of the manuscript has been rewritten (Page 5, Line 91)

3. Fig 6 and corresponding text: line 203—do you mean Fig. 6f? line 205-do you mean Fig. 6g? Fig 6 a and b are confusing. The difference in motility of $\Delta\text{flgE}::\text{Km} + \text{flgE}(\Delta\text{D3-D4})$ on the two plates is due to the length of incubation (32 vs 56 h). Where is the control of wildtype at 56h? I presume that the motility is larger. What is the significance of these differences?

Regarding the first question:

We actually meant Fig. 6g in both cases. Thank you for noticing this. It has now been corrected in the text.

Regarding the second and third questions:

The control of wild type at 56 hours is not shown, because it covered the plate. This is now mentioned in the legend for Figure 6. We wanted to show the FlgE D4 and D3 domain deletion mutant strain in motility agar at 56 h, because it is slightly motile. The fact that it is somewhat motile in motility agar is not clear from the motility agar plate incubated for only 32 hours

4. Line 392: “Figure prepared Figure prepared with Coot” needs correction.

The sentence has been corrected

5. Fig. 5: there is no legend for 5c and 5d.

The legend has been completed. Sorry about this.